# The Impact of Non-Adherence to Antihypertensive Drug Therapy

**DOI:** 10.3390/healthcare11222979

**Published:** 2023-11-18

**Authors:** Syed Karam Mustafa Gardezi, William W. Aitken, Mohammad Hashim Jilani

**Affiliations:** 1TJ Samson Community Hospital, Glasgow, KY 42141, USA; 2Department of Public Health Sciences, University of Miami Miller School of Medicine, Miami, FL 33136, USA; wwa11@miami.edu; 3Division of Cardiovascular Health, University of Cincinnati Medical Center, Cincinnati, OH 45219, USA; jilanimm@ucmail.uc.edu

**Keywords:** non-adherence, cardiovascular disease, hypertension, mortality

## Abstract

Medication non-adherence is a major healthcare barrier, especially among diseases that are largely asymptomatic, such as hypertension. The impact of poor medication adherence ranges from patient-specific adverse health outcomes to broader strains on health care system resources. The Centers for Disease Control (CDC) Wide-ranging Online Data for Epidemiologic Research (WONDER) database was used to retrieve Centers for Medicare and Medicaid Services’ data pertaining to blood pressure (BP) medication adherence, socio-economic variables, and cardiovascular (CV) outcomes across the United States. Multivariable linear regression models were used to estimate the change in total CV deaths as a function of non-adherence to BP medications. For every percent increase in the non-adherence rate, the total number of CV deaths increased by 7.13 deaths per 100,000 adults (95% CI: 6.34–7.92), even after controlling for the percentage of residents with access to insurance, the percentage of residents who were eligible for Medicaid, the percentage of residents without a college education, median home value, income inequality, and the poverty rate (*p* < 0.001). There is a significant association between non-adherence to BP medications and total CV deaths. Even a one percent increase in the adherence rate in the United States could result in tens of thousands of preventable CV deaths. Based on recently published CDC data, this could also have a tremendous impact on health care costs. This provides compelling evidence for increased efforts to improve adherence.

## 1. Introduction

Medication non-adherence is a major healthcare barrier with repercussions ranging from patient-specific adverse health outcomes to substantial strains on healthcare system resources. Adherence, in broad terms, is defined as “the extent to which a person’s behavior—taking medication, following a diet, and/or executing lifestyle changes, corresponds with agreed recommendations from a health care provider” [1]. In other words, the degree to which patients’ behaviors reflect agreed-upon plans encompasses not only medication usage, but also these other lifestyle variables. This relationship between poor adherence to medications and therapeutic lifestyle changes has been termed the “healthy adherer effect”, wherein patients who are more apt to take prescribed medications have better health outcomes above and beyond those attributable to their medications [2]. The chronicity of disease is frequently tied to increased non-adherence, with data indicating that diseases such as hypertension or diabetes, that have been present for multiple years, are complicated by medication non-adherence in 40–50% of cases [3].

Consequently, qualifying what entails proper adherence is often challenging. Major epidemiologic studies do not present unified criteria for defining adherence, resulting in variations in the reported data. However, by any standard, it is evident that medication non-adherence is exceedingly pervasive throughout the United States. A large-scale meta-analysis aiming to consolidate 569 US studies over the past few decades reported that 24.8% of individuals are non-adherent as defined by the individual studies [4]. Another meta-analysis focusing specifically on cardiovascular medications in a sample size of over 370,000 patients reported that only 57% of patients utilized at least 80% of their medications based on pharmacy refill data [5].

Social Determinants of Health (SDoH) have a well-documented relationship with medication non-adherence in the existing literature. One analysis divided over 3000 Medicaid beneficiaries into four social-risk groups, taking into account various factors such as income and education level. They sought to assess the impact of being in higher-risk groups on antihypertensive medication non-adherence. The results indicated a 36% increase in medication non-adherence in the highest-risk social group [95% CI, 0.53–0.78]. Individual SDoH in the analysis, such as income, were associated with increased non-adherence as well [6]. Another meta-analysis sought to consolidate over 29 large-scale studies to analyze the impact of SDoH on non-adherence and found a very strong association between various parameters, such as food insecurity and housing instability [7]. This very seasoned relationship between SDoH and medication non-adherence served as an inspiration for us to evaluate whether non-adherence itself could be an independent predictor of mortality for patients with hypertension, a chronic condition that is far more prevalent in populations who suffer from adverse socioeconomic variables.

Non-adherence rates are associated with far-reaching healthcare consequences that go beyond individual complications. According to the IMS Institute for Healthcare Informatics, the vast majority of adherence-related expenses occur due to avoidable hospitalizations [8]. In fact, approximately 10% of all hospitalizations in the elderly are attributed to non-adherence [9]. Other major expenses are incurred due to the worsening of otherwise treatable diseases, including increased physician and emergency department (ED) visits, escalating medication requirements, and diagnostic testing that could otherwise be avoided [10]. In 2016, the CDC Million Hearts Initiative estimated that preventable fatal and nonfatal cardiovascular events accounted for approximately 2 million hospitalizations at a cost of USD 37 billion, with projections for USD 170 billion being incurred in hospitalization-related costs between 2017 and 2022 [11]. Globally, the IMS estimates that the annual unnecessary costs related to non-adherence ranges between USD 100 billion and 300 billion, or a staggering 2–6% of global healthcare spending [8]. These expenditures are exacerbated by the effect of premature mortality, or “working years lost,” where decreased productivity costs related to preventable disease progression are estimated to be 2–3 times higher than direct healthcare costs [12].

The Million Hearts Initiative suggested that over 400,000 cardiovascular deaths (157 per 100,000) nationally were preventable in 2016. The extent to which these avoidable deaths might be ameliorated by improved medication adherence is unclear. Patients with hypertension are significantly more prone to adherence issues, largely because of the chronicity of the disease and lack of immediately observable manifestations [13]. Survey data from the National Health and Nutrition Examination Survey (NHANES) 2011–2014 showed that only 47% of individuals being treated for hypertension had their BP within range [14]. Keeping the nationwide ramifications of medication non-compliance in mind, we sought to elucidate the impact of poor blood-pressure medication adherence on healthcare outcomes, namely total cardiovascular mortality.

## 2. Materials and Methods

The CDC Atlas of Heart Disease and Stroke was used to retrieve Centers for Medicare and Medicaid Services Chronic Conditions Data Warehouse cardiovascular-disease-related data (e.g., antihypertensive medication adherence) across the United States. The CDC Wide-ranging Online Data for Epidemiologic Research (WONDER) database was used to retrieve vital statistics (e.g., mortality) and census data (e.g., educational attainment, poverty status). Multiple linear regression models were used to estimate the change in total cardiovascular deaths as a function of non-adherence to blood pressure medications.

More specifically, blood pressure medication non-adherence was defined as a proportion of days a beneficiary was covered with blood pressure medication less than 80% of the time. This was assessed using prescription drug claims data among Medicare Advantage or Medicare fee-for-service beneficiaries over 65 years of age with Medicare Part D coverage in 2015. The threshold of 80% was utilized in our research as this is the most commonly studied parameter in which existing studies have defined non-adherence and allowed for standardization of our findings with the existing literature [5]. If certain entries included missing data at any point during this period, they were excluded from analysis. Socioeconomic variables were accessed using U.S. Census Bureau American Community Survey (5-year estimate). Cardiovascular outcomes were downloaded from the Interactive Atlas of Heart Disease and Stroke, a website developed by the Centers for Disease Control and Prevention, Division for Heart Disease and Stroke Prevention [15]. Deaths were measured per 100,000 adults aged over 35 years between 2013 and 2015.

Univariable linear regression models were used to estimate the change in total cardiovascular disease (CVD) deaths as a function of non-adherence to blood pressure medications, non-adherence to diuretic medications, non-adherence to renin–angiotensin–aldosterone system (RAAS) medications, percent insured, percent eligible for Medicaid, high school education attainment, college education attainment, percent with a female head of household, percent using supplemental and nutrition benefits, median home value, median household income, income inequality, percent poverty, and unemployment. A follow-up multivariable linear regression model was estimated, and covariates for this model were selected using a combination forward–backward stepwise procedure (*p*-enter = 0.05 and *p*-exit = 0.10). Regarding model assumptions, linearity and homoscedasticity were assessed using residual plots, and normality was assessed using QQ plots. Influential outliers were assessed using box plots and Cook’s distance estimates. Regarding model fit, multicollinearity was assessed using variance inflation factors and tolerance statistics, as well as shared variance proportions.

## 3. Results

National data on prescription medication adherence, as measured by percentage of days patients have access to prescriptions in the outpatient setting, found wide-ranging disparities in adherence among disparate geographic and socioeconomic groups, as has been previously reported [10]. Controlling for all other variables in the model, including percentage of residents with access to insurance, the percentage of residents who were eligible for Medicaid, the percent of residents without a college education, the percentage of homes in the county where there was a female head of household, median home value, income inequality, and the poverty rate in the county, there were significant associations between both income inequality and blood pressure medication non-adherence and total CVD deaths; see Table 1.

There were 8.97 fewer deaths (95% CI: 5.09–12.84) per 100,000 for every standard deviation decrease in income inequality, as measured by county-level Gini coefficient, which is the summary of the dispersion of income across total income distribution. Controlling for all other variables in the model, there was also a negative association between an increase in insurance rate and the total number of CVD deaths. That is, for every percent increase in the insurance rate, the total number of CVD deaths declined by approximately 3.33 (95% CI: −3.99 to −2.67) per 100,000. Similarly, controlling for all other variables in the model, there was a negative association between increasing home value and total CVD deaths: for every one-thousand-dollar increase in the home value, the total number of CVD deaths declined by approximately 0.26 (95% CI: −0.31 to −0.21) per 100,000. Conversely, a decrease in collegiate education was associated with an increase in the total number of CVD deaths of 2.94 (95% CI: 2.44–3.43) per 100,000, and an increase in the rate of women serving as head of household was associated with an increased number of total CVD deaths of 1.38 (95% CI: 0.76–2.00) per 100,000. Further, the increasing rate of Medicaid eligibility was associated with an increased number of total CVD deaths by 0.85 (95% CI: 0.36–1.33) per 100,000, and every percent increase in poverty rate was associated with 0.86 (95% CI: 0.08–1.64) deaths per 100,000.

After controlling for all these variables, the most prominent finding was the association between non-adherence to blood pressure medications and total CVD deaths. That is, for every percent increase in the non-adherence rate, the total number of CVD deaths increased by approximately 7.13 (95% CI: 6.34–7.92) per 100,000; see Figure 1. Thus, the significant effect of non-adherence to mortality rate is demonstrated. Please find summary statistics attached in our Appendix A.

## 4. Discussion

The findings that both non-adherence and social determinants of health are independently associated with excess cardiovascular mortality are clinically meaningful and compliment the previous literature. The relationship between anti-hypertensive non-adherence and mortality, approximately 7 excess deaths per 100,000 for every one percent decrease in adherence, while unprecedented, is comparable to previously reported non-adherence studies among patients taking statins, the gold standard for cardiovascular disease prevention. A meta-analysis of 44 epidemiological studies assessing adherence to cardiovascular therapies (including anti-hypertensive and statin therapy) found a relative risk of all-cause mortality of 0.55 (0.46–0.67) and 0.71 (0.64–0.78) among patients with good medication adherence to statins and anti-hypertensive therapies, respectively [16]. These mortality differences have significant healthcare policy implications given the ubiquity of hypertension, with current estimates suggesting a prevalence of up to 45% of the adult US population [14], and preventable cardiovascular deaths, with an estimated 157 preventable CVD deaths per 100,000 people across the US [17]. These findings, in addition to the association between income inequality and mortality, suggest that economic standing, access to care, and educational attainment all play significant roles in both adherence and health outcomes. Accordingly, identifying and addressing these social determinants of health as a way to prevent non-adherence should be an area of renewed emphasis for cardiovascular disease prevention.

The difficulty of creating sustainable and scalable medication adherence programs is largely a consequence of the imperfect understanding of the underlying causes of non-adherence among patients. The World Health Organization highlights five interacting dimensions that affect adherence: patient-specific factors (e.g., poor activation, life stressors), socio-economic factors (e.g., poverty, educational attainment), healthcare system factors (e.g., access to care, insurance status), condition-related factors, and therapy-related factors [1]. While survey data have shown that forgetfulness is a commonly cited cause of non-adherence [13], it stands to reason that financial insecurity, inadequate support, and poor understanding are among the root causes of forgetfulness. Addressing these factors early in the patient–clinician relationship might strengthen the therapeutic alliance that is too often marginalized by inadequate communication, complex care systems, and time constraints. This patient-centered approach could complement other therapy-related efforts (e.g., once-daily dosing, poly-pills) and healthcare-system-related adherence programs (e.g., frequent communication between clinic visits). These same interventions might prove useful for improving adherence to the healthy lifestyle efforts that contribute to cardiovascular disease prevention, such as diet, exercise, and tobacco avoidance.

Given the clinical and societal significance of therapeutic adherence, studies have examined a wide range of interventions aimed at addressing some of these factors and improving medication adherence. One review sought to analyze the efficacy of a number of interventions [18] among a total of 182 medication-adherence randomized clinical trials. Of these, 17 focused specifically on methods for improving antihypertensive adherence. Nursing care interventions [19,20,21], pharmacist care interventions [22], simplified dosing [23,24], culturally tailored education [25], telecommunication outreach [26], and adherence incentives [27] all led to improvements in adherence to some degree. The interventions that demonstrated improved adherence generally involved multi-faceted approaches and employed tailored support from health care providers, community support (from family or friends), intensive education, and counseling. While these resource-intense interventions might seem costly, the alternative is exorbitant. Given the overwhelming financial strain of nonadherence on the healthcare system, this labor-intensive strategy would likely garner significant long-term savings. It is estimated that a multidisciplinary primary care model may cost 30% less due to decreased hospitalizations and the more appropriate use of healthcare resources [28].

While the aforementioned results are substantial, this study does have limitations. Since data were gathered from an existing database, the study design is retrospective. The secured information was provided at the county level, and so potentially confounding patient-specific factors such as health literacy were not assessed. Health literacy, or the degree to which individuals have the ability to understand and use healthcare services, has been shown to be vital in improving medication non-adherence and a decrease in health literacy has been associated with worsened patient outcomes and increased complexity of care [29]. Our dataset did not stratify between the type of antihypertensives besides diuretics and RAAS inhibitors, and thus we cannot provide further information on whether specific medications in other categories are more likely to promote non-adherence. Finally, we estimated non-adherence based on pharmacy-refill data and have not directly assessed how each individual in the study takes medications at home.

## 5. Conclusions

Our analysis of Medicare-beneficiary data across the US demonstrates a significant association between antihypertensive medication non-adherence and total CV deaths. Prior to our analysis, we already had strong evidence corroborating a relationship between negative SDoH and antihypertensive medication-nonadherence. Given that our data show that medication non-adherence appears to be an independent risk factor for CV mortality, in addition to being more prevalent in populations on the wrong side of the SDoH spectrum. This provides compelling evidence for increased efforts to improve adherence by addressing SDoH. Resolving such issues would ultimately have a tremendous impact on CV mortality. In an era where personalized medicine is increasingly tailoring treatments to patients, it is time to bring personalized medicine to healthcare delivery. To quote William Osler, “The good physician treats the disease; the great physician treats the patient who has the disease”. We can achieve this by identifying our patients’ SDoH and addressing them accordingly: educational programming for gaps in health literacy, communication and engagement, reduced costs, and a streamlined delivery.

## Figures and Tables

**Figure 1 healthcare-11-02979-f001:**
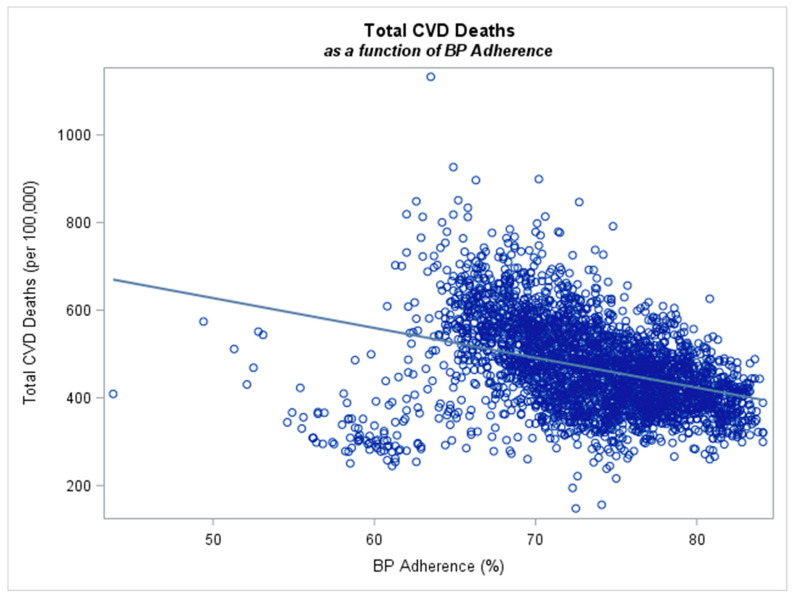
Total cardiovascular deaths as a function of anti-hypertensive adherence. For every percent increase in the adherence rate, the total number of CVD deaths decreased by approximately 7.13 (95% CI: 6.34–7.92) per 100,000 (*n* = 3192 counties).

**Table 1 healthcare-11-02979-t001:** Total CVD deaths.

	Valid N	Total CVD Deaths Beta Coefficient (95% CI)		*p*
		Unadjusted	Adjusted	
Blood Pressure Non-Adherence * (∆1%)	3192	6.82 (6.20–7.44)	7.13 (6.34–7.92)	<0.001
Diuretic Non-Adherence (∆1%)	3211	7.50 (6.80–8.21)		
RAAS Non-Adherence (∆1%)	3211	8.36 (7.52–9.20)		
Insured (∆1%)	3132	4.66 (3.99–5.33)	−3.33 (−3.99 to −2.67)	<0.001
Medicaid-Eligible (∆1%)	2915	5.95 (5.58–6.32)	0.85 (0.36–1.33)	0.001
No High School (∆1%)	3211	5.57 (5.11–6.03)		
No College (∆1%)	3211	5.90 (5.57–6.24)	2.94 (2.44–3.43)	<0.001
Female Head of Household (∆1%)	3211	4.94 (4.46–5.42)	1.38 (0.76–2.00)	<0.001
Food Benefits (∆1%)	3132	7.46 (7.09–7.83)		
Median Home Value ^†^ (∆1%)	3209	−0.61 (−0.65 to −0.57)	−0.26 (−0.31 to −0.21)	<0.001
Median Household Income ^†^ (∆1%)	3132	−4.52 (−4.76 to −4.29)		
Income Inequality ^‡^ (∆1SD)	3211	19.47 (15.62–23.31)	8.97 (5.09–12.84)	<0.001
Poverty (∆1%)	3132	8.86 (8.42–9.30)	0.86 (0.08–1.64)	0.03
Unemployment (∆1%)	3210	7.19 (5.54–8.84)		

Note: Adjusted N = 2897 counties. All unadjusted estimates are significant at *p* < 0.001. * Non-adherence defined as patients covered with medications <80% of total days. ^†^ Median Home Value and Household Income measured in thousands of $ (USD). ^‡^ Income Inequality based on the Gini coefficient, which summarizes dispersion of income across entire income distribution, with a range from 0 (perfect equality) to 1 (perfect inequality).

## Data Availability

The data underlying this article are available in a publicly accessible repository and can be obtained with the assistance of the Centers for Disease Control and Prevention (CDC) Wide-ranging Online Data for Epidemiologic Research (WONDER) database, available at: http://wonder.cdc.gov, accessed on 31 July 2021.

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
