# Peer review of "The Impact of Non-Adherence to Antihypertensive Drug Therapy"

_healthcare, 2023, doi:10.3390/healthcare11222979_

Round 1

Reviewer 1 Report

Comments and Suggestions for Authors

The manuscript highlights the importance of adherence to reduce cardiovascular complications and death. It is worth publishing, however, the are some improvements that can be made to elevate the paper.

  • The article clearly states the rationale for addressing the research problem, however, it is not necessarily novel. I believe confirmatory analyses is important to emphasise this as an important facet of healthcare studies, but it requires bolstering to address recommendations.
  • Much of the emphasis is placed on the non-adherence, which is not an unheard of association between healthcare outcomes. I recommend that associations between non-adherence and social determinants be done to bolster the results. Since the data is already available, such statistics will contribute and allow for further discussions, possibly leading to possible areas of publich health education and mitigation strategies. 
  • How was missing data accommodated? The expectation would be that much of this might be deficient in certain parameters, so it must be clearly stated.
  • The conclusion will need to be revised based on further data analysis to provide novelty to the topic.

Comments on the Quality of English Language

Minor language editing can be done as the manuscript is reviewed.

Author Response

Your feedback that our analysis could be bolstered by detailing how SDoH affect medication non-adherence is outstanding and we have indeed taken it into consideration. You astutely pointed out that we have data on how SDoH relates to mortality and we have data on how medication non-adherence relates to mortality, but not specifically on the link between SDoH and medication non-adherence. If medication non-adherence can be linked to SDoH that would significantly improve the overall message of our study.

There is tremendous amount of recent data linking SDoH to medication non-adherence in existing literature. We have added a comprehensive new paragraph in our “Introduction” section (third paragraph) detailing the strong link between SDoH and non-adherence. We included studies specifically talking about hypertension as well as large scale meta-analyses. We believe adding this association into our manuscript will significantly improve the relevance of our results.

Ultimately, we wanted our personal analysis to be focused on antihypertensive medication non-adherence and mortality and that is why it is submitted as a “Brief Report”, but with your advice we truly strengthened the impact of our investigation by linking SDoH with non-adherence. We edited and expanded our conclusion section with this information as well just as you recommended.

Finally, we appreciate you reminding us that we must make a clear statement on how missing data was accommodated and we have done so in the Material and Methods section, indicating that if we did not have consistent pharmacy refill data for entries within the dataset they were excluded.

Reviewer 2 Report

Comments and Suggestions for Authors

In general it is an interesting and well written paper that aims to raise awareness regarding the importance of antihypertensive medication adherenence.

The introduction is concrete and well referenced. The only recommendation I would think about is the addition of a reference supporting the chronicity of a disease with decreased medication adherence.

The material and Methods are presented in a satisfactory way. 

The authors supported their estimations regarding blood pressure medication non-adherence as a proportion of days a beneficiary was covered with blood pressure medication less than 80% of the time. However, this should be supported more in order to increase the reliability of the results.

The number of records that were used for the analysis is not mentioned along with the percentage of cases used for the analysis in comparison to the complete dataset for the reported period.

Moreover, differences regarding non-adherence among different antihypertensive medication categories might be interesting as well for the readers.

The discussion is on the spot and well referenced.

Regarding limitations the estimation and not direct assessment of non-adherence is a strong one and should therefore mentioned and discussed as well.

The major finding that non-adherence affects mortality rates should be emphasised in the results.

Author Response

We thank you for your excellent feedback and compliments. We have sought to address your concerns.

Your first point involved adding a reference speaking to how chronicity of disease relates to medication non-adherence. We were able to find a relevant source and insert it into our article in the introduction, finding that chronic diseases have rates of medication non-adherence up to 40-50% which is very high compared to acute disease processes.

You delve into how we came up with the number for medication non-adherence, defined based on the duration individuals were covered less than 80% of the time. We added in our material and methods section how the parameter of 80% is the most commonly cited threshold in existing literature defining non-adherence and we also implemented this so that it is uniform with said studies.  We also included a relevant source.

You raised a very interesting point that it would be useful to see if there is any variation on medication non-adherence stratified based on the specific type of antihypertensive medication. Asides from total antihypertensive mediations, the other two types of antihypertensives we have data on are RAAS inhibitors and diuretics, which we included in our table. As we do not have other data, I have added it to our limitations section. Additionally, you requested we bolster our limitation section regarding the estimations as opposed to direct assessment of medication non-adherence, and we have done so.

You asked us to reinforce the major findings that non-adherence affects mortality rates, and we have done so in our results section.

Reviewer 3 Report

Comments and Suggestions for Authors

Thank you for the opportunity to revise this manuscript.

The text deals precisely with the problem of drug therapy adherence, especially antihypertensive drugs, and the repercussions of this behavior on health outcomes.

Congratulations to the authors for the precision adopted in outlining the scientific framework of the problem and the statistical analyses. However, to improve the appearance of the manuscript, I have some observations to make:

- I found no ethical references about the conduction of the study (informed consensus, ethical conduct). Could you clarify this?

- The conclusions are consistent with the evidence and arguments presented, however the limitations of the study have not been completely described. In particular, when you write: "potentially confounding patient-specific factors were not assessed lines 209-210", it could be more precisely described. If I can suggest a phenomenon potentially related to this poor adhesion could be health literacy, that some authors describe as related to health outcomes and that can intervene as a booster phenomenon in the occurrence of any poor adhesion. In fact, HL is a potential risk factor for health outcomes (such as poor medication adherence or reduced utilisation of preventive services), and may have intervened in your population, affecting the results. I suggest reading the next recent article, that at the beginning of the background describes the phenomenon in question and could be cited for your purposes: "Cocchieri, A., Pezzullo, A. M., Cesare, M., De Rinaldis, M., Cristofori, E., & D’Agostino, F. (2023). Association between health literacy and nursing care in hospital: A retrospective study. Journal of Clinical Nursing, 00, 1–11. https://doi.org/10.1111/jocn.16899".

- The references cited are mostly not recent publications (in the last 5 years). Where and if possible, more recent references should be used.

Author Response

Thank you for your thoughtful feedback. We have sought to address each of your points.

We further bolstered our “Informed Consent” portion of our manuscript. We added that this is all de-identified data from a publicly accessible database with no risk of harm to anyone involved in the study.

Thank you very much for linking us this extremely informative article on health literacy and showing us how it can bolster our limitations section on being more specific with “patient-oriented” confounders. We added health literacy as a possible patient-specific limitation and cited the article in question as well.

We agree that some of our references could be more current. In addition to the above article, we have added 7 new references that include studies within the past 5 years. We have also removed 5 references that were particularly old.

Round 2

Reviewer 1 Report

Comments and Suggestions for Authors

The comments have been addressed, and the manuscript's discussion has been approved. The social determinants of health can be expanded on statistically in comparison to adherence, however, authors have not done so. Should the editor and co-reviewer be fine with it being absent, then it is acceptable for publication